# Effects of Gibberellic Acid on Soluble Sugar Content, Organic Acid Composition, Endogenous Hormone Levels, and Carbon Sink Strength in Shine Muscat Grapes during Berry Development Stage

Xiujie Li [1], Zhonghui Cai [2], Xueli Liu [3], Yusen Wu [1], Zhen Han [1], Guowei Yang [1], Shaoxuan Li [4], Zhaosen Xie [2], Li Liu [1,*] and Bo Li [1,*]

1   Shandong Academy of Grape, Shandong Academy of Agricultural Sciences, Jinan 250100, China; lixiujie-2007@163.com (X.L.); senwy886@163.com (Y.W.); yanhui65@163.com (Z.H.); guoweiyang66@163.com (G.Y.)
2   College of Horticulture and Garden, Yangzhou University, Yangzhou 225009, China; caizhonghui2023@163.com (Z.C.); xiezhaosen@yzu.edu.cn (Z.X.)
3   Taishan Institute of Science and Technology, Taian 271000, China; lily0538@163.com
4   Qingdao Academy of Agricultural Sciences, Qingdao 266000, China; lsxsdau@163.com
*   Correspondence: gssliuli@shandong.cn (L.L.); sdtalibo@163.com (B.L.)

**Abstract:** The phytohormone gibberellic acid ($GA_3$) is widely used in the table grape industry. However, there is a paucity of information concerning the effects of $GA_3$ on fruit quality and sink strength. This study investigated the effects of exogenous $GA_3$ treatments (elongating cluster + seedless + expanding, T1; seedless + expanding, T2; expanding, T3; and water, CK) on the content of sugars, organic acids, and endogenous hormones and sink strength. Results showed that T2 treatment displayed the highest fructose and glucose levels at 100 days after treatment (DAT), whereas its effect on tartaric acid, malic acid, and citric acid concentrations at 80 and 100 DAT was relatively weak. Under $GA_3$ treatments, $GA_3$, IAA, and CTK contents increased, whereas ABA content decreased at 1, 2, 4, 8, and 48 h. Analysis of sugar phloem unloading revealed that T2 treatment exhibited the highest values during softening and ripening stages. Our findings indicate that appropriate $GA_3$ application can positively influence sink strength by regulating sink size and activity, including berry size enlargement, sugar phloem unloading, and sugar accumulation in grape sink cells.

**Keywords:** gibberellin; grape; soluble sugar; organic acid; sugar unloading; sink strength

## 1. Introduction

Gibberellin, a pivotal phytohormone that interacts synergistically with other plant hormones, functions as a central regulator of diverse developmental processes within plants. Gibberellic acid 3 ($GA_3$), a specific form of gibberellin, plays a critical role in promoting parthenocarpy and has garnered extensive attention for application in the cultivation of seedless fruits, augmentation of berry dimensions, and elongation of rachis across a wide spectrum of grape cultivars [1–4]. $GA_3$, in conjunction with auxin, plays a pivotal role in stimulating cell division and expansion. This interaction regulates fruit development and subsequent enlargement following fertilization [5]. Recently, the consumption of fruits has increased, owing to their high internal and external appearance quality. This trend is associated with economic growth and changing consumer preferences. To meet market demands, producing grapes with standardized cluster length and uniform and large berry size has become crucial [6]. Consequently, gibberellin finds widespread application in production.

Presently, a number of studies have demonstrated the efficacy of gibberellin as an effective agent in enhancing the elongation of grape inflorescence. Sun [7] reported that

the application of gibberellin to Cabernet Franc grapes led to a proportional increase in peduncle length with an increase in gibberellin concentration. Similarly, Wang et al. [8] observed that grape varieties such as Midnight Beauty, Zaoheibao, and Summer Black exhibited elongated fruit pedicels following gibberellin treatments at a concentration of 10 mg/kg. Yang et al. [6] revealed that the application of exogenous $GA_3$ not only inhibited the synthesis of endogenous gibberellin, but also orchestrated the modulation of gibberellin signal transduction, thereby promoting inflorescence elongation.

Prior research has highlighted the utilization of $GA_3$ as a viable strategy in grape cultivation, particularly for seed management. Notably, the application of $GA_3$ solution at a concentration of 100 mg $L^{-1}$ prior to full bloom has demonstrated its potential to produce seedless cultivars and prompt seed abortion in seeded cultivars [1]. Han and Lee [9] observed that $GA_3$ exhibited efficacy in not only fostering fruit enlargement, but also augmenting cluster length, cluster weight, and berry weight. Korkutal et al. [4] further substantiated the multifaceted impacts of gibberellins, revealing their role in promoting stem elongation, triggering flowering induction, stimulating pollen tube growth, yielding seedless fruits, and increasing the size of seedless berries. Zhao [10] expounded upon the benefits of employing $GA_3$ and streptomycin before the full-bloom stage, coupled with $GA_3$ application alone post this stage. This approach was found to notably increase berry size of "Shenxiangwuhe" and "12–17", concurrently facilitating seedlessness in the latter. For the grape variety "Zhuosexiang", the combined administration of $GA_3$ and forchlorfenuron (CPPU) before full bloom, followed by $GA_3$ application post full bloom, exhibited comparable benefits, yielding a remarkable seedless rate of 71%. Notably, $GA_3$ application emerged as the most effective approach in enhancing single berry weight and seedless rate, as underscored by Zhao et al. [10].

Sink strength, which has been elucidated as the competitive capacity of an organ to efficiently intake photoassimilates, can be considered as a product of two critical components: sink size and sink activity [11]. Sink size pertains to the physical limitation represented by the total biomass of the sink organ. Notably, the quantification of cells, both in terms of their number and size in the sink organ, can serve as a suitable metric for evaluating sink size. Sink activity is regarded as the physiological limitation that governs the import of assimilates into a sink organ. It is determined by three pivotal physiological processes. Firstly, it involves the unloading of assimilates from the phloem, followed by the subsequent transport of sugars beyond the phloem, leading to their absorption by the sink organ. Secondly, the sink organ's own respiratory consumption contributes to sink activity. Thirdly, the accumulation of carbohydrates within sink organs also influences sink activity [12]. Prior research has indicated that phytohormones can modulate sink strength. Notably, hormones such as $GA_3$, cytokinin (CTK), abscisic acid (ABA), and auxin have been recognized as participants in this regulatory process [13,14]. However, the underlying mechanism through which GA specifically influences sink strength in perennial fruit crops, including grapes, remains elusive.

Shine Muscat is a prominent table grape variety extensively cultivated in China. This variety is widely known for its excellent characteristics, such as large size, sweet taste, good storage capacity, and aromatic qualities; however, certain issues, such as the occurrence of seeded fruit, bitter seeds and pericarp, and tightly packed bunches, remain persistent. GA is widely employed to cultivate larger, seedless berries and enhance overall fruit quality. Previous investigations have demonstrated that GA can induce inflorescence elongation, foster the development of seedless fruits, and enhance the size of seedless berries in different grape varieties. Our previous research has also indicated that GA influences the aromatic components in Shine Muscat berries [15]. However, a research gap remains, particularly in understanding the interrelationship among the contents of soluble sugars, organic acids, and endogenous hormones and sink strength in Shine Muscat grapes subjected to $GA_3$ treatments. Therefore, the present study aimed to investigate how $GA_3$ influences the contents of soluble sugars and organic acids, preliminarily explore the interrelationship

among these components, and clarify the mechanism underlying the fruit sink strength modulation by $GA_3$.

## 2. Materials and Methods

### 2.1. Field Conditions and Materials

The experiment was conducted at the Jinniushan vineyard in Tai'an, Shandong, China (36.127° N, 117.004° E). For the study, Shine Muscat (*Vitis labruscana*) grapes, which were self-rooted and planted in 2012, were used. The planting arrangement involved a plant density of 3.0 m² (with a spacing of 1.0 m between plants and 3.0 m between rows) within a rain shelter cultivation system. For the experimentation, a total of 30 healthy and well-developed vines were meticulously selected, with each treatment group comprising three replicates. Randomly selected branches with moderate vigor and a consistent number of leaves were chosen on the vines where one inflorescence on each branch was left and selected as the test object. Each inflorescence was trimmed one week before flowering to comprise only 5 cm of the apex. Trimmed inflorescences were assigned to one of three treatments or one control. The vines were subjected to $GA_3$ (ProGibb 40, Valent BioScience, Walnut Creek, CA, USA) treatment at varying concentrations and during distinct periods. The experimentation comprised four distinct treatment groups, including three $GA_3$ treatment groups and one control (CK) group, as detailed in Table 1.

**Table 1.** Experimental design of different gibberellin treatments.

| No. | Treatment | Note |
|-----|-----------|------|
| CK | Water | 1. Rachis elongation treatment: twenty days prior to anthesis, inflorescences were immersed in a solution containing 5 mg/L $GA_3$ for 5 s.<br>2. Seedlessness treatment: a day after full bloom, inflorescences were immersed in a solution containing 25 mg/L $GA_3$ for 5 s.<br>3. Expansion treatment: within two weeks after bloom withering, inflorescences were immersed in a solution containing 30 mg/L $GA_3$ for 5 s.<br>4. CK treatment: inflorescences were treated with water (treated at the same time as T1). |
| T1 | Rachis elongation treatment + Seedlessness treatment + Expansion treatment | |
| T2 | Seedlessness treatment + Expansion treatment | |
| T3 | Expansion treatment | |

### 2.2. Tissue Collection

For each treatment, 30 berries were randomly chosen from 30 vines at 7:00 am every 14 days [14, 28, 42, 56, 70, 84, and 98 days after treatment (DAT)] for analysis of longitudinal diameter (LD) and transverse diameter (TD). For each treatment, 120 berries were randomly chosen from 30 vines on a regular basis at 7:00 am every 20 days [20, 40, 60, 80, and 100 days after treatment (DAT)] post expansion treatment. The samples were segregated into two groups. The samples from the first group, comprising 60 berries from each treatment, were collected and transported in an ice box to the laboratory. The berries were then analyzed for berry weight, soluble solids (SS), and titratable acidity (TA). The samples from the second group, comprising 60 berries from each treatment, were frozen in liquid nitrogen and immediately stored in a refrigerator at −80 °C. These samples were intended for the assessment of soluble sugars (glucose, fructose, and sucrose) and organic acids (tartaric acid, citric acid, and malic acid) in the grape berries. Moreover, samples comprising 30 berries with consistent fruit diameter, obtained from 30 berries of each treatment, were collected at specific intervals: 0, 1, 2, 4, 8, 24, and 48 h post expansion treatment. Following liquid nitrogen treatment, these samples were enveloped in tinfoil, refrigerated, and subsequently brought back to the laboratory. The samples were then stored in a low-temperature refrigerator (−80 °C) for the determination of indole-3-acetic acid (IAA), CTK, GA, and ABA contents in grape berries within one month.

### 2.3. Measurements of Berry Growth

The weight of individual fruits was determined using an electronic balance, with an accuracy of 0.01 g. For each analysis, a total of 30 berries were randomly selected from 30 clusters and each of the three replications. LD and TD of the fruit were measured using a vernier caliper. SS was assessed by utilizing an aliquot of grapevine juice with the aid of a digital refractometer (Automatic Refractometer SMART-1, Atago, Tokyo, Japan), which is denoted as Brix. Total acidity was quantified by an ATAGO (PAL-1) hand-held digital refractometer.

### 2.4. High-Performance Liquid Chromatography Analysis of Glucose, Fructose, and Sucrose

Sugars were extracted following the procedure: for each treatment, 1 g of homogenized grape berry material was accurately weighed and then diluted to 10.0 mL using ultrapure water (Millipore, Bedford, MA, USA). The solution was subsequently incubated for 20 min in a water bath set at 35 °C. Afterward, the supernatant was subjected to centrifugation at $21,000 \times g$ for 10 min at room temperature (BHG-Hermle Z 365, Wehingen, Germany). The extraction process was repeated three times, and the resulting supernatants were pooled. The liquid supernatant underwent filtration through a 0.22 μm, 13 mm diameter syringe filter (Shanghai Xingya Purification Material Factory, Shanghai, China). Subsequently, the filtered solution was employed for the analysis of glucose, sucrose, and fructose contents. High-performance liquid chromatography (HPLC; Waters, Milford, MA, USA) was employed for the analysis. The separation conditions used for the soluble sugar analysis were as follows: detector, differential refractive index detector (RID); column, YMC-Pack Polyamine II (4.6 mm × 250 mm); phase, acetonitrile/water = 75:25 (*v/v*); flow rate, 0.8 mL/min; injection amount, 10 μL; column temperature, 40 °C; analysis time, 20 min. The eluted peaks were detected utilizing an RID-10A differential refractive index detector (Shimadzu Co., Ltd., Kyoto, Japan). The quantification of glucose, fructose, and sucrose was conducted using standard curves of authentic compounds. Each treatment was replicated three times.

### 2.5. High-Performance Liquid Chromatography Analysis of Tartaric Acid, Citric Acid, and Malic Acid

Organic acid analyses were conducted utilizing a Waters series 515 chromatography unit, which was equipped with two 515 pumps and a 2487 dual UV detector (Waters Alliance 2695 HPLC) operating at a wavelength of 210 nm. The separation conditions used for organic acid analysis were as follows: column, Thermo Hypersil COLD aQ (4.6 mm × 250 mm, 5 μm); phase, 10 mmol/L $NH_4H_2PO_4$ (pH = 2.3)/methanol = 98/2 (*v/v*); flow rate, 0.8 mL/min; injection amount, 10 μL; column temperature, 25 °C; analysis time, 20 min. Quantification of tartaric acid, citric acid, and malic acid was performed using standard curves of authentic compounds. The analysis encompassed extracts from three replicate tissue samples.

### 2.6. Extraction and Determination of $GA_3$, IAA, CTK, and ABA

HPLC (Nexera LC-30A, Shimadzu, Japan) was employed for analysis of $GA_3$, IAA, CTK, and ABA. For each treatment, 5 g of grape berry was ground to powder under liquid nitrogen. After extraction, 50 mL of 80% (*v/v*) MeOH (methanol) and 50 μL of 30 mg/mL sodium diethyldithiocarbamate were added. After full oscillation, samples were kept at 0 °C overnight. After filtration, the residue was washed twice with 40 mL and 20 mL 80% methanol, respectively. The filtrate was dried at 40 °C on a rotary evaporator. The distillation bottle was flushed with 10 mL of petroleum ether and 10 mL of phosphoric acid buffer twice, and the flushed solution was passed through a 0.45 μm filter membrane. The water phase was decolorized 3 times with petroleum ether (equal volume). The pH value was adjusted to pH = 8, and the samples were extracted with ethyl acetate (equal volume) 3 times. Then, when pH was adjusted to pH = 3, the water phase was extracted with ethyl acetate (equal volume) 3 times, and the extract was dried at 40 °C. The extract was

dissolved with 1 mL of 50% MeOH, filtered with 0.45 μm filter membrane, and 20 μL of the sample was taken for HPLC analysis. The standard hormone sample that was utilized was a product of Sigma Company. The chromatographic conditions were as follows: mobile phase, methanol: 0.8% glacial acetic acid solution = 55:45 (*v/v*); flow rate, 0.8 mL/min; injection amount, 10 μL; column temperature, 30 °C; detection wavelength, 254 nm. The analysis comprised extracts from three replicate tissue samples.

*2.7. Analysis of Sugar Phloem Unloading*

The sugar phloem unloading was assessed using the fruit cup method [16]. During the softening stage (August 19) and ripening stage (September 28), one sunward-facing grain in grape berries was randomly selected for each treatment. A cross of about 1 cm was delicately marked at the grape's umbilical region using a scalpel. This process ensured that the flesh remained unharmed and the grape flesh tissue was not cut. Subsequently, the peel was gently separated from the pedicel using tweezers and carefully cut with anatomical scissors. The prepared Mes buffer, comprising 5 mmol/L Mes (2-(n-morph) ethanesulfonic acid), 2 mol/L $CaCl_2$, 100 mmol/L mannitol (D-mannitol), and 0.2% (*w/v*) polyvinylpyrrolidone (PVPP), was then used to rinse the peel. The peeled grape fruit was meticulously placed into a 20 mm infusion tube filled with Mes buffer. The opening was sealed with a film, and the fruit cup was secured on the cluster. The outer layer of the fruit cup was covered with tin foil to prevent temperature fluctuations from affecting the test results and extending the "berry cup's" longevity. Sampling times were designated as 9:30–10:00, 11:30–12:00, 13:30–14:00, 15:30–16:00, and 17:30–18:00, during which the replaced buffer was discharged from the cup. Each time, the buffer liquid from the fruit cup was collected, and it was filled using a syringe. The collected samples were immediately frozen using liquid nitrogen and stored at −80 °C for subsequent analysis of glucose, fructose, and sucrose.

*2.8. Statistical Analysis*

Data analysis was performed using Microsoft Excel 2010 and SPSS 26.0. Graphical representations were created using Prism 9. Pearson correlation analysis was conducted using R statistical software (Version 3.0.3).

## 3. Results

*3.1. Effects of $GA_3$ on Physicochemical Characteristics of Grape Berries*

To investigate the effects of $GA_3$ on Shine Muscat grapes, various key attributes of grape berries during development were evaluated in both $GA_3$-treated and CK groups. These parameters included berry weight, berry size, LD, TD, vertical/horizontal ratio (a measure of fruit shape index), TA (g/L), and SS (Brix) of the berries. As illustrated in Figure 1A, the trend of changes in berry weight was analogous between $GA_3$-treated groups and the CK group. Notably, at each developmental stage, significantly higher berry weights were observed under T1, T2, and T3 treatments than those under the CK treatment. Furthermore, berry weights were substantially higher under T1 and T2 treatments than those under the T3 treatment. At 20–100 DAT, the berry size exhibited a comparable trend to the berry weight for both $GA_3$-treated groups and the CK group. The berry weight and volume were more rapidly increased under the T2 treatment than those under any other treatment (as shown in Figure 1A,B). Specifically, the berry weights were 11.3 g and 12.05 g in 2022 and 2021, respectively, marking a remarkable 52.2% (2022) and 42.8% (2021) increase, respectively, over those in the CK group at 100 DAT (Figure 1A). Similarly, the berry volumes were 10.59 $cm^3$ and 10.63 $cm^3$ in 2022 and 2021, respectively, representing a substantial 21.8% (2021) and 59.2% (2022) increase, respectively, compared to those in the CK group at 100 DAT (Figure 1B). Moreover, bunch compactness decreased in response to $GA_3$ treatments (T2 and T3 treatments). The berry size was significantly higher in T1, T2, and T3 treatment groups than in the CK group (Figure 1C), which suggests an expansion effect of $GA_3$.

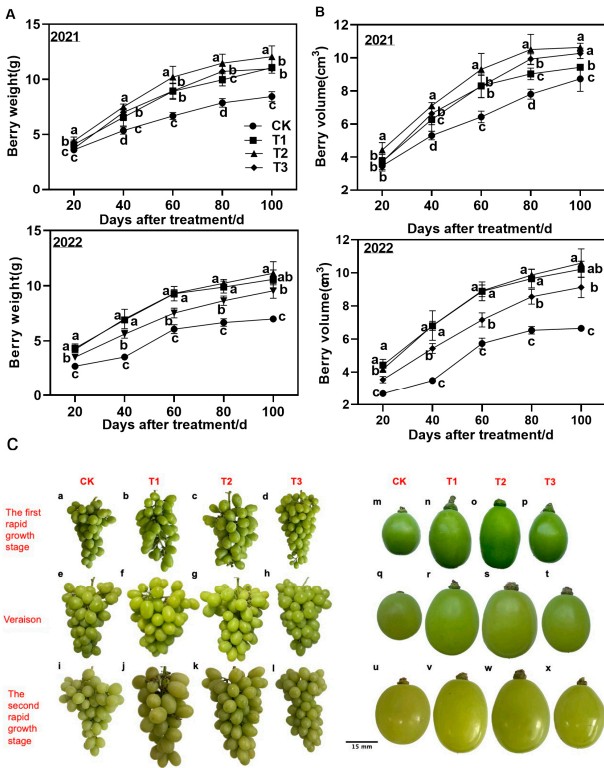

**Figure 1.** Berry weight (**A**) and volume (**B**) in 2021 and 2022, and berry growth at three stages: hard-core stage, softening stage, and harvest stage (**C**) of "Shine Muscat" under various GA$_3$ treatments. a–x: berry corresponding to respective treatments and periods. Different letters denote statistically significant differences among treatments at the same period, as determined by Duncan's test ($p < 0.05$).

Moreover, no significant increase in TD was observed between GA$_3$-treated groups and the CK group (Figure 2A). In contrast, LD was substantially higher in the three GA$_3$-treated groups than in the CK group at 20–100 DAT (Figure 2B). However, similar LD values were observed for T1 and T2 treatment groups. The fruit shape index was evidently higher in T1 and T2 treatment groups than in the other groups, with that in the CK group being the lowest. At 21–70 DAT, a slight increase in fruit shape index in the Tl group was evident compared to that in the T2 group (Figure 2C).

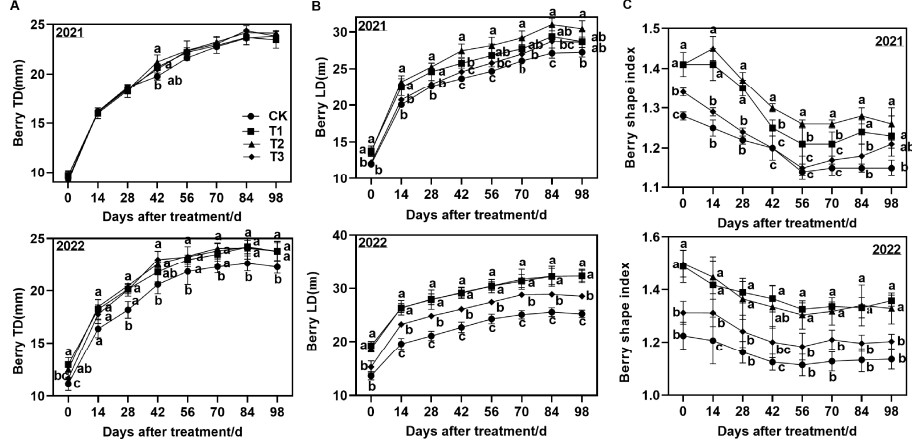

**Figure 2.** Transverse diameter (**A**), longitudinal diameter (**B**), and berry shape index (**C**) in 2021 and 2022 of "Shine Muscat" under various GA$_3$ treatments. Different letters denote statistically significant differences among treatments at the same period, as determined by Duncan's test ($p < 0.05$).

The SS content in grape berries is illustrated in Figure 3A. It displayed an increasing trend as the fruits ripened. In 2021, at 20 DAT, remarkably higher SS contents were observed in GA-treated groups than those in the CK group. At 40 DAT, the SS content demonstrated the following trend: T2 > T3 > T1 > CK. At 60, 80, and 100 DAT, significantly higher SS contents were observed under T1 and T2 treatments than those under T3 and CK treatments. Moreover, in 2022, SS contents increased across all treatments as the fruits matured. At 60 DAT, a significant difference in SS content was observed between the T3 and CK groups. However, at 20, 40, 80, and 100 DAT, no significant difference was observed between GA-treated groups and the CK group.

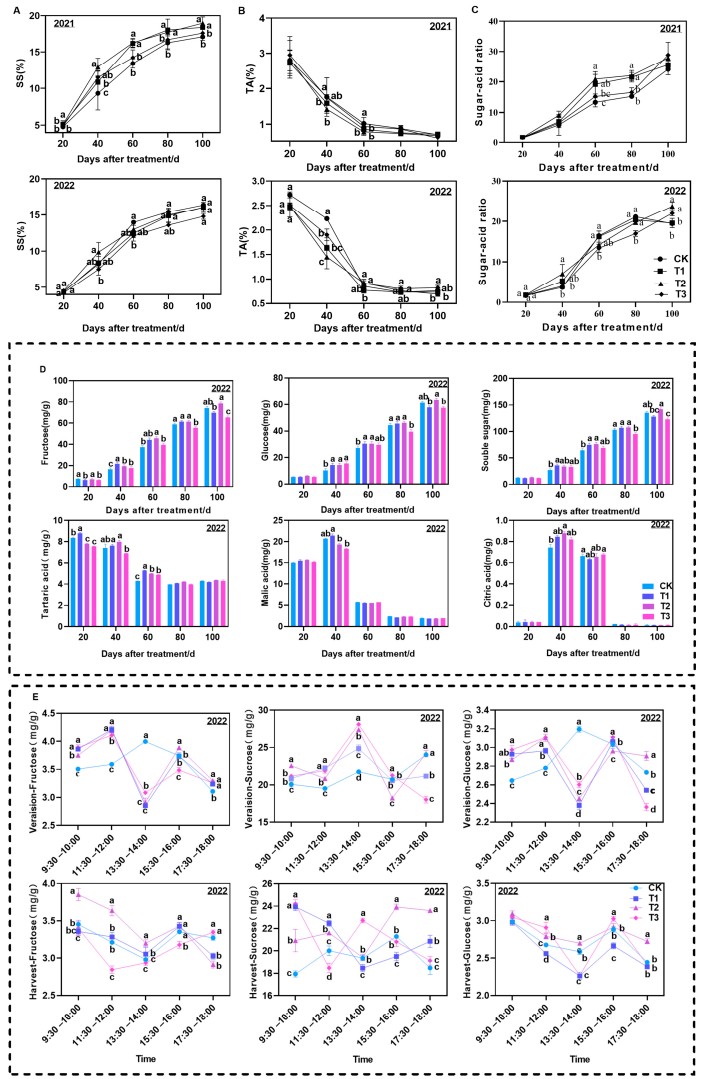

**Figure 3.** Soluble solid (SS) content (**A**), titratable acidity (TA) (**B**), and SS/TA ratio (**C**) in 2021 and 2022. Contents of soluble sugars and organic acids (**D**), and sugar phloem unloading (**E**) in "Shine Muscat" grapes during veraison and harvest stages under various gibberellin treatments in "Shine Muscat" grapes. Different letters denote statistically significant differences among treatments at the same period, as determined by Duncan's test ($p < 0.05$).

An evident declining trend in TA during fruit development was observed, particularly at 20–60 DAT (Figure 3B). In 2021, no significant differences were observed among the treatments at 20, 80, and 100 DAT. However, at 40 DAT, CK and T3 groups exhibited higher TA compared to T1 and T2 groups. At 60 DAT, the CK group displayed the highest TA among the treatments. In 2022, no significant differences in TA were recorded among the treatments at 20 DAT. At 40 DAT, the overall trend observed was as follows:

CK > T3 > T1 > T2, with TA in the CK group being significantly higher than that in GA-treated groups. At 100 DAT, a significant difference in TA was observed under the T3 and T1 (or T2) treatments.

The SS/TA ratio demonstrated a consistent increasing trend during the course of fruit development. In 2021, varying GA treatments yielded minimal influence on the SS/TA ratio across different time periods. However, at 100 DAT, the SS/TA ratio in CK, T1, and T2 groups was significantly higher than that in the T3 group; however, no significant differences were observed among CK, T1, and T2 groups (Figure 3C).

### 3.2. Effect of $GA_3$ on Soluble Sugar Content in Grape Berries

As illustrated in Figure 3D, the soluble sugar composition of grape berries primarily consisted of glucose and fructose. Notably, the sucrose content was negligible. The glucose and fructose contents within all four groups exhibited a consistent increasing trend; however, throughout the hanging life, the fructose content consistently surpassed that of glucose. The increase in total sugar content was significantly higher at 40–60 DAT than during the maturation stage (80–100 DAT). Prior to reaching 60 DAT, the total sugar content in T1 and T2 treatment groups was significantly higher than that in the CK group. Moreover, the total sugar content in the T3 group was the lowest, at 80 DAT, among all four treatment groups. During the ripening stage, the T2 group exhibited the highest total sugar content among all four treatment groups.

The fructose and glucose contents within each treatment group gradually increased throughout the hanging life of the fruit. Notably, the changes in glucose and fructose contents in fruits at 80 DAT closely paralleled those observed in the total sugar content. Upon attaining complete maturation, the contents of fructose and glucose in the T2 treatment group exceeded those observed in other treatment groups (glucose: 78.66 ± 1.30 mg/g; fructose: 63.49 ± 1.03 mg/g). Notably, among all groups, the lowest glucose (65.36 ± 1.40 mg/g) and fructose (57.58 ± 1.01 mg/g) contents were observed in the T3 treatment group.

### 3.3. Effect of $GA_3$ on Organic Acid Content in Grape Berries

As illustrated in Figure 3D, the primary organic acids present in grape berries included tartaric acid, malic acid, and citric acid, with citric acid content being relatively minimal. During the period of hanging life, we noted a distinctive V-shaped trend in the tartaric acid content, which peaked at 20 DAT and plummeted at 80 DAT across all treatments. Prior to 60 DAT, significant differences were observed in the tartaric content among all four treatments; however, these differences were not evident at 100 DAT. At 100 DAT, we observed the lowest and highest tartaric acid contents under T1 (4.31 ± 0.08 mg/g) and T2 (4.38 ± 0.02 mg/g) treatments, respectively.

The malic acid content in grape berries initially increased and then decreased over the course of hanging life. Notably, a significant difference among the four treatments was evident at 40 DAT alone. During the maturation phase, the lowest malic acid content was observed under the T1 treatment (1.83 mg/g), which was 8.74% lower than that observed under the CK treatment.

The change trends in citric acid content paralleled those of malic acid content. At 40 DAT, the citric acid content in all groups reached its peak during the hanging life. Notably, significant differences in citric acid content were observed across all groups at 40 and 60 DAT. At 100 DAT, the lowest citric acid content was noted under the T2 treatment (0.012 mg/g), which was 8.33% lower than those observed under the T1 and T3 treatments.

### 3.4. Analysis of Sugar Unloading in Phloem

Our results demonstrated that the primary sugars in grape berries were fructose and glucose, with relatively minimal sucrose content at the veraison stage. Notably, the fructose content exceeded that of glucose. As depicted in Figure 3E, the changes in fructose and glucose unloading in the phloem exhibited a double-peak curve at five distinct time points following $GA_3$ treatment. In the morning, sugar unloading increased rapidly, reaching

its peak at 12:00, closely aligned with the increase in photosynthetic rate. Subsequently, a sharp decline was observed at 14:00. Furthermore, a resurgence in sugar unloading was observed with an increase in photosynthetic rate, reaching a minor peak at 16:00, followed by a gradual decline at sunset. Notably, the morning unloading volume was approximately 1.1 times that of the afternoon unloading. However, the changes in fructose and glucose unloading in the phloem exhibited a unimodal curve at the five time points under the CK treatment. In the morning, sugar unloading increased rapidly, reaching its peak at 14:00, followed by a rapid decrement.

The maximum fructose unloading (3.89 mg/g) was observed between 9:30 and 10:00 in the T3 treatment fruits, significantly surpassing that observed in the CK treatment fruits (3.51 mg/g). For the T1 treatment, the peak fructose unloading (4.21 mg/g) occurred between 11:30 and 12:00. Similarly, the peak fructose unloading for the CK treatment reached 4.00 mg/g at 13:30–14:00 and that for the T2 treatment reached 3.89 mg/g and 3.30 mg/g at 15:30–16:00 and 17:30–18:00, respectively.

Furthermore, the peak glucose unloading for the T3 treatment occurred at 9:30–10:00 and 11:30–12:00, measuring 2.98 mg/g and 3.11 mg/g, respectively. For the CK treatment, the highest glucose unloading occurred at 13:30–14:00, following a pattern similar to that of fructose. Notably, the T3 treatment exhibited the highest glucose unloading (3.11 mg/g) at 15:30–16:00. In addition, at 17:30–18:00, glucose unloading was significantly higher in the T2 group than that observed in other groups. Overall, these findings indicate that the highest sugar (fructose+glucose) unloading (32.25 mg/g) was observed in the T2 treatment group during the veraison stage.

Similarly, in vivo analysis of sugar unloading at the maturation stage resembled that at the veraison stage. The primary sugars in grape berries were fructose and glucose, with relatively minimal amounts of sucrose owing to incomplete transformation. As illustrated in Figure 3E, the changes in fructose and glucose unloading in the phloem exhibited a unimodal curve at each time point during the maturation stage. Sugar unloading reached its peak at 10:00 in the morning, gradually declining to a minimum at 14:00. Subsequently, sugar unloading increased again with an increase in photosynthetic rate, achieving a minor peak at 16:00 and, finally, decreasing at sunset. The unloading value observed in the morning was one fold higher than that observed in the afternoon. Among the treatments, the T2 treatment demonstrated the highest fructose unloading at the first four time points, while the highest unloading under the T3 treatment was observed at 17:30–18:00. Moreover, the maximum glucose unloading under the T2 treatment was evident at 9:30–10:00 (3.09 mg/g), 13:30–14:00 (2.92 mg/g), and 17:30–18:00 (2.72 mg/g). The T3 treatment group exhibited the highest glucose unloading at 11:30–12:00 (2.91 mg/g) and 15:30–16:00 (3.03 mg/g).

Based on the sugar unloading data at various time points for each treatment, the T2 treatment displayed the highest sugar unloading (fructose + glucose) at the maturation stage, with a value of 31.26 mg/g.

*3.5. Effect of GA$_3$ on Endogenous Hormone Contents in Grape Berries*

Four types of endogenous hormones—GA, ABA, IAA, and CTK—were analyzed. The GA content demonstrated a fluctuating trend at 0–48 h after GA$_3$ expansion treatment. It reached a peak at 0–4 h, followed by a sharp decline and a rapid increase at 24 h. In contrast, the GA content under the CK treatment exhibited a gradual increase at 48 h. Notably, the overall trend in GA content at 1–24 h was as follows: T3 > T2 > T1 > CK (Figure 4A). The ABA contents were significantly higher in CK treatment fruits than those in fruits subjected to the GA treatment. The lowest and highest ABA contents in berries were observed at 4 h and 24 h after GA treatment, respectively. At 48 h, the overall trend in ABA content was as follows: CK > T1 > T3 > T2 (Figure 4B). The IAA content in fruits displayed a trend similar to that observed for GA. The highest IAA content in berries under the T1 treatment was observed at 8 h, and at 48 h, the following overall trend was noted: T3 > T2 > T1 > CK (Figure 4C). The CTK content in berries increased rapidly at 0–2 h under CK, T2, and T3

treatments, followed by a slower upward trend that gradually decreased at 4 h. Moreover, the CTK content showed an upward trend under the T1 treatment at 0–8 h. At 48 h, the following trend was observed: T2 > T3 > T1 > CK (Figure 4D).

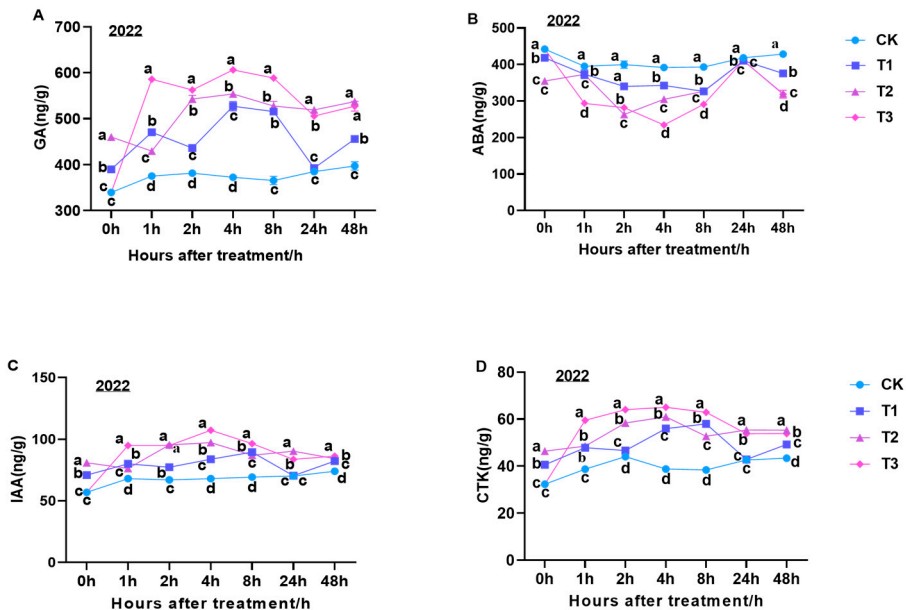

**Figure 4.** Contents of endogenous hormones—gibberellic acid (GA₃) (**A**), abscisic acid (ABA) (**B**), indole-3-acetic acid (IAA) (**C**), and cytokinin (CTK) (**D**)—in "Shine Muscat" grapes under various GA₃ treatments. Different letters denote statistically significant differences among treatments at the same period, as determined by Duncan's test ($p < 0.05$).

### 3.6. Correlation Analysis

The correlation analysis depicted in Figure 5 examines the relationships among the contents of endogenous GA, soluble sugars, organic acids, and key parameters, namely, berry weight, berry shape index, TD, LD, volume, SS, and TA. Through correlation analysis, we observed a significant positive correlation between the GA content and the key parameters, including berry weight, berry shape index, TD, LD, and volume, particularly with TD. Furthermore, a significant negative correlation was noted between GA and SS contents at 60, 80, and 100 DAT, as well as between the GA content and TA at 20 and 40 DAT. In contrast, a significant positive correlation was observed between the GA content and TA at 60 and 80 DAT (Figure 5A).

The contents of reducing sugars, glucose, and fructose in fruits displayed a significant positive correlation with GA contents at 40 and 60 DAT (Figure 5B); however, a significant negative correlation between them was observed at 80 and 100 DAT. Notably, the GA content was significantly negatively correlated with tartaric acid content at 20 DAT; malic acid content at 40, 60, 80, and 100 DAT; and citric acid content at 80 and 100 DAT (Figure 5B). In contrast, the GA content was significantly positively correlated with tartaric acid content at 60, 80, and 100 DAT; malic acid content at 20 DAT; and citric acid content at 20, 40, and 60 DAT (Figure 5B).

The heatmap presented in Figure 5C indicates that IAA and CTK contents were significantly positively correlated with GA₃ contents at 1, 2, 4, 8, 24, and 48 h. Furthermore, the correlation analysis highlighted that ABA contents were significantly negatively correlated with GA₃ contents and IAA and CTK contents at 1, 2, 4, 8, 24, and 48 h. Notably, a significant positive correlation was observed between IAA and CTK contents.

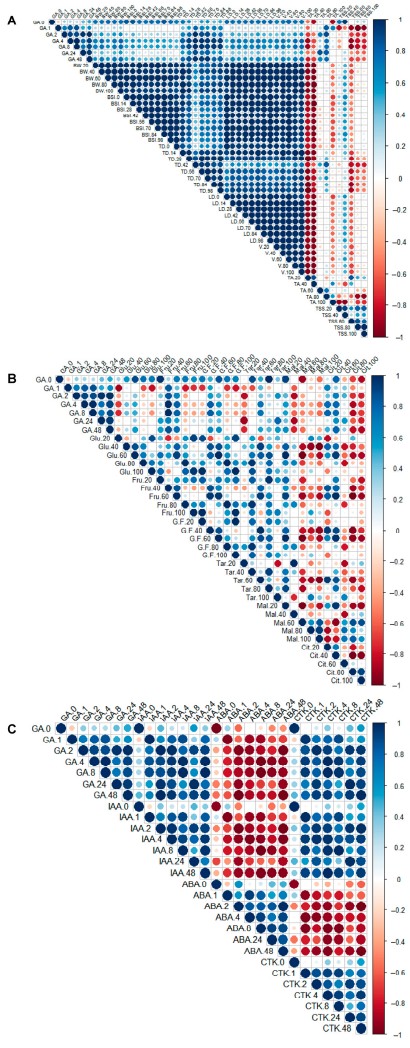

**Figure 5.** Correlation analysis of GA$_3$ content with key parameters (**A**), soluble sugar and organic acid contents (**B**), and endogenous hormone contents (**C**) from grapes under various GA$_3$ treatments.

## 4. Discussion

### 4.1. GA$_3$ Application by Cluster Dipping at Different Periods Results in Berry Size Enlargement in Grapes

Our study conclusively showed that GA$_3$ effectively enhanced the size of Shine Muscat berries. Intriguingly, we found that the effect of GA$_3$ on LD was more significant than that on TD. Subsequently, we conducted additional analyses to measure berry weight and volume in order to investigate the enlargement effect. As anticipated, GA$_3$ significantly enhanced both berry weight and volume, particularly under the seedlessness + expansion treatment (T2 treatment). This observation potentially indicates that the concentrations of compounds, such as sugars and acids, might have been regulated through expansion dilution.

### 4.2. GA$_3$ Application Alters the Concentrations of Sugars and Acids

To verify the hypothesis mentioned earlier, we conducted analyses of soluble sugar and organic acid contents. The application of GA led to a significant increase in SS content in berries during 2021, which is consistent with the findings of Tyagi et al. [17] in Sangiovese grapes at 79 d. However, this effect was not observed during 2022, consistent with previous findings for Sangiovese grapes at 49 d [17]. We further observed lower TA in GA-treated berries than that in the untreated control at 40 and 60 DAT in 2021 and 40 DAT in 2022. This finding aligns with that of Gao et al. [2] in Cabernet Sauvignon grapes. However, no significant difference in TA was observed at 80 and 100 DAT in

2021 and 80 DAT in 2022, aligning with the findings of previous research on Sangiovese grapes [17]. In commercial settings, the SS/TA ratio is considered the most reliable indicator of fruit flavor [18]. This study demonstrates that GA-treated berries [rachis elongation (28/4/2022) + seedlessness (24/5/2022) + expansion treatment (7/6/2022), or seedlessness (24/5/2022) + expansion treatment (7/6/2022)] exhibited a higher SS/TA ratio at 80 DAT in both 2021 and 2022, suggesting improved berry flavor. Subsequently, we investigated the effect of GA application on soluble sugar and organic acid contents in fruits to thoroughly explore the potential existence of the expansion dilution effect.

It is widely accepted that soluble sugars in grape berries primarily consist of glucose and fructose. Although some studies have reported the presence of sucrose in certain table grape varieties [19], sucrose has not been detected in Cabernet Sauvignon grapes [6]. In the present study, traces of sucrose were detectable, albeit in limited quantities. Monosaccharides serve as effective osmotica in plants. Herein, GA treatment [seedlessness (24/5/2022) + expansion treatment (7/6/2022)] increased the accumulation of monosaccharides (fructose and glucose) in grape berries at 100 DAT. However, berries treated with expansion alone exhibited the lowest values, suggesting an effect of expansion dilution. This phenomenon may be influenced by other phytohormones mediated by GA or from the competition for photosynthates due to seed development. Our findings indicate that appropriate $GA_3$ application (T2 treatment) can enhance sugar accumulation, overcoming the potential impact of expansion dilution.

### 4.3. Interactions between Signaling Pathways of $GA_3$ and Other Phytohormones Contribute to the Coordinated Development of Berries

GA, in conjunction with IAA and CTK, plays a role in promoting cell division and expansion, leading to increased cell growth and protein synthesis [20]. Therefore, these hormones coordinate the development and enlargement of fruits following fertilization. Notably, these three plant hormones, along with ABA, have significant regulatory functions in the establishment and maintenance of fruit sink strength [13]. The application of exogenous hormones could potentially trigger changes in endogenous hormone levels in plants [21]. Therefore, in this study, we assessed alterations in the levels of four endogenous hormones ($GA_3$, CTK, IAA, and ABA). Our findings indicate that $GA_3$ application led to increased levels of IAA and CTK compared with those in the CK group, while also exerting an inhibitory effect on ABA production. The correlation analysis unveiled a significant positive correlation between $GA_3$ and IAA/CTK contents and a significant negative correlation between $GA_3$ and ABA contents. These results, in conjunction with prior research, suggest a potential crosstalk between GA and other plant hormones, such as auxin and CTK [22]. Interestingly, the growth of grape berries induced by auxin and CK treatment is also partially dependent on GA biosynthesis [23], underscoring the significance of the synergy between these plant hormones. $GA_3$ has been substantiated to enhance fruit size by promoting cell division and expansion, contributing to the mechanisms of cell wall formation and relaxation [24]. Previous studies have revealed that auxin modulates grape berry ripening, which could be associated to cell expansion [25,26]. Numerous studies have also provided support for the notion that CTK could govern critical rate-limiting steps in nutrient distribution and utilization [27,28]. These findings collectively suggest the significant contributions of GA, IAA, and CTK in modulating fruit sink strength.

### 4.4. $GA_3$ Application Increases the Sink Strength in Grapes

The application of gibberellin can lead to an increase in auxin content. Gibberellin, along with auxin and CTK, plays a role in nutrient transport and attraction, ensuring that the fruit becomes a robust "reservoir" and is positioned favorably in the nutritional competition. Sink organs in plants serve as net importers of assimilates. Throughout various stages of plant development, all plant organs can function as sinks, thereby receiving assimilates. In terms of the transport of assimilates, the capacity of a sink organ to import assimilates is referred to as its sink strength. However, a substantial proportion of the imported

assimilates can be utilized for respiratory processes within the sink organs. Consequently, measuring the sink strength of a sink organ solely based on parameters such as the absolute growth rate or net accumulation rate of dry matter falls short of accurately gauging the actual capacity of a sink organ to receive assimilates. This conventional approach instead represents an apparent sink strength. Measuring the import rate of assimilates, which takes into account the sum of the net carbon gain and the carbon loss through respiration within a sink organ, would provide a more accurate estimate of the true sink strength. In addition, CTK has the potential to enhance sink capacity by promoting cell proliferation or sustaining sink activity through the regulation of sucrolytic enzymes, thereby allowing the acquisition of more photoassimilates.

Sink strength is generally defined as the competitive ability of an organ to attract assimilates, represented by the product of sink size and sink activity [29]. In various fruit crops, including grapes, GA has been shown to increase fruit size by stimulating cell division and elongation [30]. Our study demonstrated that GA$_3$ application significantly increased berry size, berry weight, LD, and TD. This suggests that GA$_3$ can bolster sink strength by actively contributing to the enhancement of sink size, encompassing both cell number and size. This augmentation, in turn, ensures the accrual of a greater amount of photoassimilates and dry matter within the sink structure. Similar findings were observed in pear (*Pyrus pyrifolia*), where GA treatment led to a significant increase in sink demand and fruit size [31]. The transport of photosynthates occurs through a permeation-driven loading or unloading process within the phloem, which is governed by the pressure gradient that exists between the source and the phloem reservoir. Our findings demonstrate that an appropriate application of GA$_3$ can expedite phloem unloading, resulting in enhanced sugar unloading in berry fruits during both the veraison and maturation stages. This enhancement implies an increase in sink activity. Apart from sugar translocation, the accumulation of sugars stands as a pivotal determinant of sink strength. Our study shows that GA$_3$ application leads to the accumulation of substantial concentrations of glucose and fructose in berry fruits. This aligns with the findings of previous studies [32,33], which collectively support the notion that GA$_3$ can affect sink strength by expediting the accumulation of hexose sugars. These findings imply that the phytohormone GA$_3$ actively participates in the regulation of sink size and activity, encompassing an increase in berry size, sugar phloem unloading, and sugar accumulation in the sink cells. Altogether, these mechanisms significantly contribute to the modulation of sink strength in grapes (as depicted in Figure 6).

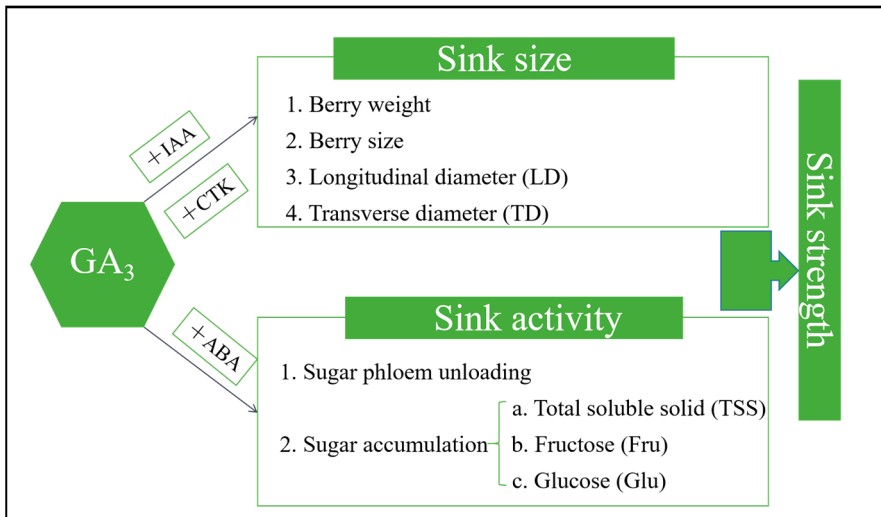

**Figure 6.** Mechanism through which GA$_3$ regulates sink strength via modulation of sink size and activity, encompassing an increase in berry size, sugar phloem unloading, and sugar accumulation in the sink cells in grapes.

## 5. Conclusions

Collectively, this study introduced two significant and novel findings that contribute to our understanding of grape physiology and growth regulation: (i) GA$_3$ exerted significant effects on the contents of soluble sugars, organic acids, and endogenous hormones (IAA, CTK, and ABA). The application of GA$_3$ led to enhanced sugar unloading during the softening and ripening stages. (ii) The appropriate application of GA$_3$ plays a crucial role in orchestrating the modulation of sink size and activity, including the enhancement of berry size, the facilitation of sugar phloem unloading, and the accumulation of sugars within sink cells. These insights collectively exert a robust effect on the overall sink strength in grape development. Moreover, these novel findings significantly enhance our comprehension of the complex interplay between GA$_3$ and soluble sugar, organic acid, and endogenous hormone contents. This study presents compelling empirical evidence that contributes to the broader body of knowledge supporting effective strategies for table grape cultivation and the proficient utilization of GA$_3$, ultimately benefiting the grape industry in China.

**Author Contributions:** Conceptualization, X.L. (Xiujie Li), Z.X., L.L. and B.L.; methodology, X.L. (Xiujie Li), Z.C. and Z.X.; sample preparation and analysis, Z.C.; investigation, X.L. (Xueli Liu) and G.Y.; data curation, Z.C. and S.L.; writing—original draft preparation, L.L.; writing—review and editing, B.L.; Supervision, X.L. (Xueli Liu), Y.W. and Z.H.; project administration, B.L.; funding acquisition, Z.H., L.L. and B.L. All authors have read and agreed to the published version of the manuscript.

**Funding:** This research was funded by the Natural Science Foundation of Shandong Province (Grant No. ZR2023MC101, ZR2021QC165, and ZR2022QC185), Agricultural Science and Technology Innovation Project of Shandong Academy of Agricultural Sciences (Grant No. CXGC2023A47), Agricultural Variety Project of Shandong Province—New Varieties Grape Cultivation of High Quality with Characteristic and Industrialization Technology Innovation and Promotion (Grant No. 2022LZGCQY019), Collaborative Extension Project of Major Agricultural Technology of Shandong Province (Grant No. SDNYXTTG-2023-18), Key R&D Program of Shandong Province (Grant No. 2022TZXD0011), and National Natural Science Foundation of China (32202462).

**Data Availability Statement:** The original contributions presented in the study are included in the article, further inquiries can be directed to the corresponding authors.

**Conflicts of Interest:** The authors declare no conflicts of interest.

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
