# Peer review of "Effects of Gibberellic Acid on Soluble Sugar Content, Organic Acid Composition, Endogenous Hormone Levels, and Carbon Sink Strength in Shine Muscat Grapes during Berry Development Stage"

_horticulturae, doi:10.3390/horticulturae10040346_

Round 1
Reviewer 1 Report
Comments and Suggestions for Authors
I found the paper interesting, I annotated minor points directly in the PDF
Figures need to be improved

Comments on the Quality of English Language
EN is of good quality
Author Response
|
Response to Reviewer 1 Comments
|
||
|
1. Summary |
|
|
|
Thank you very much for taking the time to review this manuscript. Please find the detailed responses below and the corresponding revisions/corrections highlighted/in track changes in the re-submitted files.
|
||
|
2. Questions for General Evaluation |
Reviewer’s Evaluation |
Response and Revisions |
|
Does the introduction provide sufficient background and include all relevant references? |
Yes |
|
|
Are all the cited references relevant to the research? |
Yes |
|
|
Is the research design appropriate? |
Yes |
|
|
Are the methods adequately described? |
Yes |
|
|
Are the results clearly presented? |
Must be improved |
We have revised the manuscript according to the comments. |
|
Are the conclusions supported by the results? |
Yes |
|
|
3. Point-by-point response to Comments and Suggestions for Authors |
||
|
1. Comments 1: GA3 -not defined,never start a sentence with an acronym. |
||
|
Response 1: Thank you for pointing this out. We agree with this comment. Therefore, I/we have revised it as Gibberellic acid 3 (GA3), a specific form of gibberellin. [page 1, paragraph 1 and line 30] |
||
|
2. Comments 2: Fertilization, a supporting reference would be required. |
||
|
Response 2: Agree. A reference has been added-He, H., Yamamuro, C., (2022), Interplays between auxin and GA signaling coordinate early fruit development,Horticulture Research,9: uhab078. [page 1, paragraph 1 and line 35] 3. Comments 3: Recently, the consumption of fruits has increased owing to their high internal and external appearance quality. This trend is associated with economic growth and changing consumer preferences. To meet market demands, producing grapes with standardized cluster length and uniform and large berry size has become crucial. references? |
||
|
Response 3: Agree. A reference has been added-Yang, L.L., Niu, Z.Z., Wei, J.G.,Zhao, Y.Z., Chen. Z., Xuan, L.F., Chu, F.J., (2020), Effects of Exogenous GA3 on Inflorescence Elongation and Gibberellin Anabolism Related Gene Expression in Grapevine, Molecular Plant Breeding, 18:5600-5606. [page 1, paragraph 1 and line 39]. |
||
|
4. Comments 4: Furthermore, this research seeks to contribute substantial data to support the cultivation of table grapes and the utilization of GA in China. this statement would reduce the interest of an internatoinal journal....delete? |
||
|
Response 4: Agree. We have deleted this sentence. [page 1, paragraph 1 and line 40]. |
||
|
5. Comments 5: “3.0 per hectare” please check...only 3? p/ha? |
||
|
Response 5: Agree. The planting arrangement involved a plant density of 3.0 m2(with a spacing of 1.0 m between plants and 3.0 m between rows) . [page 3, paragraph 1 and line 103] |
||
|
6. Comments 6: Note: these are relevant info, please add this as column in Tab. 1 |
||
|
Response 6: Agree. We have added this “Note” as column in Tab. 1. [page 3, paragraph 1 and line 114]. |
||
|
7. Comments7: ‘The seeds were then analyzed for berry weight’ seeds---for berry? please rephrase. |
||
|
Response 7: Agree. It should be “ berry”, or “fruit grain”, not seeds.[page 3, paragraph 1 and line 120]. |
||
|
8. Comments8: LD and TD not defined |
||
|
Response 8: LD and TD has been defined in the previous paragraph-The berries were then analyzed for berry weight, longitudinal diameter (LD), transverse diameter (TD), total soluble solids (TSS), and titratable acidity (TA). [page 3, paragraph 1 and line 121]. |
||
|
9. Comments 9: Fruits should be fruit. |
||
|
Response 9: Agree. We have revised it. [page 4, paragraph 1 and line 136]. |
||
|
10. Comments 10: Figure 1. very small panels, please split the figure to improve its readability |
||
|
Response 10: Agree. Thank you for this suggestion. We have split the Figure and now it is clear. [page 7, paragraph 1 and line 242]. |
||
|
11. Comments 11: Figure 4 these figures are not easy to read |
||
|
Response 11: Agree. Figure 4 shows the correlation analysis among the contents of endogenous GA, soluble sugars, organic acids, and key parameters, namely, berry weight, berry shape index, TD, LD, volume, SS, and TA.[page 11, paragraph 1 and line 382]. |
||
|
12. Comments12: ‘more significant’ how did you compare significance? |
||
|
Response 12: Agree. No significant increase in TD was observed between GA3-treated groups and the CK group (Figure 3A). In contrast, LD was substantially higher in the three GA3-treated groups than in the CK group at 20–100 DAT (Figure 3B). Therefore, the effect of GA3 on LD was more significant than that on TD. [page 7, paragraph 1 and line 242]. |
||
|
13. Comments13: GA -GA3 |
||
|
Response 13: Agree. We have revised it. [page 14, paragraph 2 and line 420]. |
||
|
4. Response to Comments on the Quality of English Language |
||
|
Point 1: Minor editing of English language required. |
||
|
Response 1: We have revised the manuscript according to the comments. |
||
|
5. Additional clarifications |
||
|
No. |
||

Reviewer 2 Report
Comments and Suggestions for Authors
The manuscript by Li et al. explores the response of Shine Muscat grapes to treatments with AG during various stages of berry development. This topic is highly significant for understanding plant biology and establishing agronomic management practices in vineyards, and it has been extensively studied in other table and wine cultivars.
However, the methodology of the work lacks clarity due to a lack of information about the experiment's description. For example, while the plantation distance is mentioned, the plant density of "3.0 per hectare" is unclear (Ln106). Additionally, the trellis system of the plants is ambiguous; it is mentioned that four shoots were trained, but it's unclear if each plant was pruned to develop only four shoots, which would imply four bunches per plant. This is inconsistent with the sampling details provided.
The treatments involve immersing inflorescences and clusters in a solution of AG3, but the duration of such immersion is not specified. Furthermore, in Ln 127, it mentions "The cluster length of 30 berries for each treatment was assessed...," but it's unclear whether this refers to the length of the berries or the length of the bunch.
Regarding seed selection, it's mentioned that seeds were selected from 30 berries at different times (Ln 129), but the evaluation of the seeds (Ln 133) appears to focus on parameters related to berry size rather than seed size.
Throughout the description, the number of berries collected does not correspond with a plant with only four shoots/bunches: 120 (Ln129) + 150 (Ln140) + 30 (Ln 147).
In Figure 1, the graphics lack indications of the different treatments, and the letters in the picture lack references. Additionally, the time is presented as "Days after treatment/d," but the dates of the treatments are different (Notes of Table 1), and there is no information on the date of the CK treatment.
Considering these writing problems, it is challenging to follow the experiment and gain a clear understanding of the obtained results.
Author Response
|
Response to Reviewer 2 Comments
|
||
|
1. Summary |
|
|
|
Thank you very much for taking the time to review this manuscript. Please find the detailed responses below and the corresponding revisions/corrections highlighted/in track changes in the re-submitted files.
|
||
|
2. Questions for General Evaluation |
Reviewer’s Evaluation |
Response and Revisions |
|
Does the introduction provide sufficient background and include all relevant references? |
Can be improved |
We have added several relevant references in the revised manuscript. |
|
Are all the cited references relevant to the research? |
Can be improved |
We have checked all cited references. |
|
Is the research design appropriate? |
Must be improved |
The experiment design has been improved. |
|
Are the methods adequately described? |
Must be improved |
Some methods have been revised. |
|
Are the results clearly presented? |
Must be improved |
We have revised the manuscript according to the comments. |
|
Are the conclusions supported by the results? |
Not applicable |
We have revised the manuscript according to the comments. |
|
3. Point-by-point response to Comments and Suggestions for Authors |
||
|
1. Comments 1: However, the methodology of the work lacks clarity due to a lack of information about the experiment's description. For example, while the plantation distance is mentioned, the plant density of "3.0 per hectare" is unclear (Ln106). Additionally, the trellis system of the plants is ambiguous; it is mentioned that four shoots were trained, but it's unclear if each plant was pruned to develop only four shoots, which would imply four bunches per plant. This is inconsistent with the sampling details provided.
|
||
|
Response 1: “3.0 per hectare” is not correct. The planting arrangement involved a plant density of 3.0 m2(with a spacing of 1.0 m between plants and 3.0 m between rows) . Thirty healthy grapevines were selected. Randomly selected branches with moderate vigor and a consistent number of leaves were chosen on the vines where one inflorescence on each branch was left and selected as the test object. Each inflorescence was trimmed one week before flowering to comprise only 5 cm of the apex. Trimmed inflorescences were assigned to one of three treatments or one control. Three GA3 treatment schemes were applied to the trimmed inflorescences (Table 1). In treatment 1 (T1), GA3 was applied approximately 20 d before bloom when the rachis was elongating, 1 d after full bloom, and 14 d after bloom. In treatment 2 (T2), GA3 was applied 1 d after full bloom and 14 d after bloom. In treatment 3 (T3), GA3 was applied on 14 d after bloom. The control check (CK) was dipped in water at the same time as T1. [page 2, paragraph 1 and line 103,106-109]. |
||
|
2. Comments 2: The treatments involve immersing inflorescences and clusters in a solution of GA3, but the duration of such immersion is not specified. Furthermore, in Ln 127, it mentions "The cluster length of 30 berries for each treatment was assessed...," but it's unclear whether this refers to the length of the berries or the length of the bunch. |
||
|
Response 2: Agree. The duration of such immersion is 5 seconds. [page 3, and line 114] "The cluster length of 30 berries for each treatment was assessed...," We have deleted this sentence, because this result of cluster length was not exhibited in this paper. I’m sorry for this mistake. |
||
|
3.Comments 3: Regarding seed selection, it's mentioned that seeds were selected from 30 berries at different times (Ln 129), but the evaluation of the seeds (Ln 133) appears to focus on parameters related to berry size rather than seed size. |
||
|
Response 3: Agree. 120 berries were randomly chosen from 30 grape clusters. [page 3, paragraph 2, and line 116-117]
|
||
|
4.Comments 4: Throughout the description, the number of berries collected does not correspond with a plant with only four shoots/bunches: 120 (Ln129) + 150 (Ln140) + 30 (Ln 147) |
||
|
Response 4: Agree. Thirty healthy grapevines were selected. Randomly selected branches with moderate vigor and a consistent number of leaves were chosen on the vines where one inflorescence on each branch was left and selected as the test object. Each inflorescence was trimmed one week before flowering to comprise only 5 cm of the apex. Trimmed inflorescences were assigned to one of three treatments or one control. Each grapevine contain four treatments (CK, T1, T2, and T3), not a plant with only four shoots/bunches. [page 3, paragraph 1, and line 106-110].
|
||
|
5.Comments 5: In Figure 1, the graphics lack indications of the different treatments, and the letters in the picture lack references. Additionally, the time is presented as "Days after treatment/d," but the dates of the treatments are different (Notes of Table 1), and there is no information on the date of the CK treatment. |
||
|
Response 5: Agree. We have added indications of the different treatments in figure1. "Days after treatment/d" means “the beginning is after expansion treatment. GA3 was applied approximately 20 d before bloom when the rachis was elongating, 1 d after full bloom, and 14 d after bloom.The control check (CK) was dipped in water at the same time as T1. [page 6, paragraph 2, and line 228]
|
||
|
6. Response to Comments on the Quality of English Language |
||
|
Point 1: I am not qualified to assess the quality of English in this paper. |
||
|
Response 1: TopEdit (www.topeditsci.com) has polished this paper during the preparation of this manuscript. |
||
|
7. Additional clarifications |
||
|
No |
||

Reviewer 3 Report
Comments and Suggestions for Authors
Authors must comply with the corrections suggested in the manuscript

Author Response
For research article
|
Response to Reviewer 3 Comments
|
||
|
1. Summary |
|
|
|
Thank you very much for taking the time to review this manuscript. Please find the detailed responses below and the corresponding revisions/corrections highlighted/in track changes in the re-submitted files.
|
||
|
2. Questions for General Evaluation |
Reviewer’s Evaluation |
Response and Revisions |
|
Does the introduction provide sufficient background and include all relevant references? |
Must be improved |
We have added several relevant references in the revised manuscript. |
|
Are all the cited references relevant to the research? |
Must be improved |
We have added several relevant references in the revised manuscript. |
|
Is the research design appropriate? |
Yes |
|
|
Are the methods adequately described? |
Must be improved |
Some methods have been revised. |
|
Are the results clearly presented? |
Can be improved |
We have revised the manuscript according to the comments. |
|
Are the conclusions supported by the results? |
Yes |
|
|
3. Point-by-point response to Comments and Suggestions for Authors |
||
|
1. Comments 1: ‘GA3, in conjunction with auxin, plays a pivotal role in stimulating cell division and expansion. This interaction regulates fruit development and subsequent enlargement following fertilization. ’Include references here
|
||
|
Response 1: Agree. A reference has been added-He, H., Yamamuro, C., (2022), Interplays between auxin and GA signaling coordinate early fruit development,Horticulture Research,9: uhab078. [page 1, paragraph 1 and line 35]
|
||
|
2. Comments 2: ‘First, it involves the unloading of assimilates from the phloem, followed by75 the subsequent transport of sugars beyond the phloem, leading to their absorption by the sink organ. Second, the sink organ’s own respiratory consumption contributes to sink activity. Third, the accumulation of carbohydrates within sink organs also influences sink activity ’It would be more appropriate to insert references here
|
||
|
Response 2: Agree. A reference has been added-Roitsch, T. (1999). Source-sink regulation by sugar and stress. Current opinion in plant biology, 2(3), 198–206. [page 2, paragraph 2 and line 78]
|
||
|
3. Comments 3 Augment -enhance
|
||
|
Response 3: Agree. We have revised it. [page 2, paragraph 3 and line 89]
|
||
|
4. Comments 4 ‘Vitis labruscana’ I think it is Vitis labrusca
|
||
|
Response 4: Vitis labruscana is correct.
|
||
|
5. Comments5 ‘3.0 per hectare’3333 trees per hectare
|
||
|
Response 5: Agree. The planting arrangement involved a plant density of 3.0 m2(with a spacing of 1.0 m between plants and 3.0 m between rows) [page 3, paragraph 1 and line 103]
|
||
|
6. Comments 6 Very confusing, you need to make the experimental design very clear. Was the experiment completely randomized or randomized blocks? What is the number of repetitions? How many observation units per plot? Were the units of observation plants or branches?
|
||
|
Response 6: Agree. Thirty healthy grapevines were selected. Randomly selected branches with moderate vigor and a consistent number of leaves were chosen on the vines where one inflorescence on each branch was left and selected as the test object. Each inflorescence was trimmed one week before flowering to comprise only 5 cm of the apex. Trimmed inflorescences were assigned to one of three treatments or one control. Each grapevine contain four treatments (CK, T1, T2, and T3). [page 3, paragraph 1, and line 106-110].
|
||
|
7. Comments 7 The values are expressed as grams of citric acid per kilogram of fresh weight. For grapes, the correct option is to transform the data into the equivalent of tartaric acid, not citric acid.
|
||
|
Response 7: Agree. I’m sorry for this mistake. Total acidity was quantified by an ATAGO (PAL-1) handed digital refractometer, which was measured by a postgraduate student. [page 4, paragraph 2, and line 139-140].
|
||
|
8. Comments 8 Analysis of sugar phloem unloading-An illustrative figure would be desirable
|
||
|
Response 8: Agree. An illustrative figure has been added. [page 5, paragraph 2, and line 197-200].
|
||
|
9. Comments 9 Figure 1.-It is mandatory to relate the graph curves with the respective treatments, as shown in figure 2A
|
||
|
Response 9: Agree. We have added indications of the different treatments in figure1, as shown in figure2A. [page 6, and line228].
|
||
|
10. Comments 10 Only SS (Soluble Solids). The term Total Soluble Solids (TSS) is no longer used |
||
|
Response 10: Agree. We have revised this.
|
||
|
4. Response to Comments on the Quality of English Language |
||
|
Point 1: I am not qualified to assess the quality of English in this paper. |
||
|
Response 1: TopEdit (www.topeditsci.com) has polished this paper during the preparation of this manuscript. |
||
|
5. Additional clarifications |
||
|
No |
||

Round 2
Reviewer 2 Report
Comments and Suggestions for Authors
I appreciate the author's effort to address the identified flaws in the manuscript. However, several points are not clear at all throughout the methodology of the work.
The training system of the plants and the pruning level and crop load are not specified. According to the interpretation of the methodology, it seems that each plant has at least 30 clusters. Is this correct? (Ln117)
The authors indicate that they selected 30 plants, but according to the experimental setup (4 treatments with three replicates), it should be 12. How many plants were analyzed? (Ln105-106). How were the plants distributed in the field?
They indicate that the total acidity was measured with a refractometer which, according to the manufacturer's specifications, does not have that capability.(Ln 140-141)
In the HPLC analysis, they indicate that each treatment was replicated three times. Does this refer to biological replicates or technical replicates? (Ln160)
There is no mention of how the berries were processed for hormone analysis. It is indicated that the berries were collected in liquid nitrogen, stored at -80°C, and then the run parameters are given. But there is no information about how the samples were processed, which makes it impossible to interpret the results.(Ln 172-178).
The experimental setup for sugar flux determination is not adequately cited. The original citation is https://doi.org/10.1093/aob/mcg159. Figure 1 is taken from that publication without reference. The bibliography presents a reference that is not found in any database. How are the results of sugar flux expressed? The unit in the graphs is mg/g, while in the mentioned work, the quantities are in the order of ug/g. (Ln179-197).
Regarding the presentation of the results, some of them are very difficult to understand, which raises significant doubts about the work. Firstly, Figure 3 shows the berry size analysis at 0, 14, 28, 42, 56, 70, 84, and 96 days after treatment. However, in the Materials and Methods section, it is mentioned that those samples were collected at 20, 40, 60, 80, and 100 days.(Ln 117-123)
The definition of Figures 2 and 3 is low; the symbols identifying each treatment cannot be visualized well, making it difficult to understand the graphs.
The representation of Figure 2C is not clear, as it shows photos of berries and clusters without explanation.
In the analysis of berry sugar composition, they indicate the presence of traces of sucrose, but they do not present the data.This does not correspond to the phloem loading data, where the predominant sugar is sucrose. There is no explanation provided for this discrepancy. Moreover, this latter finding (the high amount of sucrose in the phloem flow) is in stark contrast to the study by Wang et al that describes the experimental setup, where sucrose is very low.
On more than one occasion, evolutions throughout shelf life are mentioned(Ln 281 & 290), but there is no mention of post-harvest analysis in the Materials and Methods section.
More inconsistencies are found as the manuscript is analyzed, but at this point, it is sufficient to demonstrate that the manuscript, while interesting, does not merit publication in its current form. It should be thoroughly rewritten and reconsidered as a new submission.
Author Response
|
Response to Reviewer 2(Round 2) Comments
|
||
|
1. Summary |
|
|
|
Thank you very much for taking the time to review this manuscript. Please find the detailed responses below and the corresponding revisions/corrections highlighted/in track changes in the re-submitted files.
|
||
|
2. Questions for General Evaluation |
Reviewer’s Evaluation |
Response and Revisions |
|
Does the introduction provide sufficient background and include all relevant references? |
Must be improved |
We have added several relevant references in the revised manuscript. |
|
Are all the cited references relevant to the research? |
Must be improved |
We have added several relevant references in the revised manuscript. |
|
Is the research design appropriate? |
Yes |
|
|
Are the methods adequately described? |
Must be improved |
Some methods have been revised. |
|
Are the results clearly presented? |
Can be improved |
We have revised the manuscript according to the comments. |
|
Are the conclusions supported by the results? |
Yes |
|
|
3. Point-by-point response to Comments and Suggestions for Authors |
||
|
Comments 1: The training system of the plants and the pruning level and crop load are not specified. According to the interpretation of the methodology, it seems that each plant has at least 30 clusters. Is this correct? (Ln117)
|
||
|
Response 1: No. Each plant has at least 30 clusters is wrong. In this above text (Ln104-109), the experiment consists 30 plants, and each plant consists four treatments (T1, T2, T3 and CK). For each treatment, one branch (consisting one inflorescence ) was selected as the experimental materials. “For each treatment, 120 berries were randomly chosen from 30 grape clusters on a regular basis” means “For each treatment, 120 berries were randomly chosen from 30 plants (grape clusters ) on a regular basis”.[page 3, paragraph 2 and line 116-119]
|
||
|
Comments 2: The authors indicate that they selected 30 plants, but according to the experimental setup (4 treatments with three replicates), it should be 12. How many plants were analyzed? (Ln105-106). How were the plants distributed in the field?
|
||
|
Response 2: The experiment consists 30 plants, and each plant consists four treatments (T1, T2, T3 and CK). For each treatment, one branch (consisting one inflorescence ) was selected as the experimental materials. The planting arrangement involved a plant density of 3.0 m2(with a spacing of 1.0 m between plants and 3.0 m between rows) within a rain shelter cultivation system. [page 3, paragraph 1 and line 106-109]
|
||
|
Comments 3 They indicate that the total acidity was measured with a refractometer which, according to the manufacturer's specifications, does not have that capability.(Ln 140-141)
|
||
|
Response 3: Total acidity was quantified by an ATAGO handed digital refractometer. It has the capability for measuring the total acidity as follow:acid can be detected.
|
||
|
Comments 4 In the HPLC analysis, they indicate that each treatment was replicated three times. Does this refer to biological replicates or technical replicates? (Ln160)
|
||
|
Response 4: Each treatment consists 3 biological replicates,each biological replicates consists of 3 technical replicates.
|
||
|
Comments5 There is no mention of how the berries were processed for hormone analysis. It is indicated that the berries were collected in liquid nitrogen, stored at -80°C, and then the run parameters are given. But there is no information about how the samples were processed, which makes it impossible to interpret the results.(Ln 172-178).
|
||
|
Response 5: For each treatment, 5 g of grape berry was ground to powder under liquid nitrogen. After extraction, 50 mL of 80% (vol/vol) MeOH (methanol) and 50 µl of 30mg/ml sodium diethyldithiocarbamate were added. After full oscillation, samples were kept at 0 °C for overnight. After filtration, the residue was washed twice with 40 ml and 20ml 80% methanol, respectively. The filtrate is dried at 40°C on a rotary evaporator. Flush the distillation bottle with 10ml of petroleum ether and 10ml of phosphoric acid buffer for 2 times, and pass the flush solution through 0.45um filter membrane. The water phase was decolorized 3 times with petroleum ether (equal volume ). Adjust the pH=8, and the samples were extracted with ethyl acetate (equal volume) for 3 times. Then adjust the pH=3, the water phase was extracted with ethyl acetate (equal volume) for 3 times, and the extract was dried at 40°C. Dissolve the extract with 1ml of 50% MeOH, filtered with 0.45um filter membrane, and take 20ul of the sample for analysis. [page 5, paragraph 1 and line 175-187]
|
||
|
Comments 6 The experimental setup for sugar flux determination is not adequately cited. The original citation is https://doi.org/10.1093/aob/mcg159. Figure 1 is taken from that publication without reference. The bibliography presents a reference that is not found in any database. How are the results of sugar flux expressed? The unit in the graphs is mg/g, while in the mentioned work, the quantities are in the order of ug/g. (Ln179-197).
|
||
|
Response 6: Figure 1 is taken from “Wang, Z.P., Carbonneau, A., Deloir, A.(2006). A new method for consecutively measuring the accumulation of sugar in grape fruit. Journal of Fruit Science, 2006, 23: 770-773. https://doi.10.13925/j.cnki.gsxb.2006.05.028. Now, the total text was uploaded. In our work, the unit of glucose and fructose is mg/g, and the unit of sucrose is ug/g. In the reference, the quantities are in the order of ug/g. This maybe the result of different varieties, different treatments and different periods. [page 6, paragraph 1, and line 211].
|
||
|
Comments 7 Regarding the presentation of the results, some of them are very difficult to understand, which raises significant doubts about the work. Firstly, Figure 3 shows the berry size analysis at 0, 14, 28, 42, 56, 70, 84, and 96 days after treatment. However, in the Materials and Methods section, it is mentioned that those samples were collected at 20, 40, 60, 80, and 100 days.(Ln 117-123)
|
||
|
Response 7: I’m sorry for this mistake. In the Materials and Methods section, It should be-- For each treatment, 30 berries were randomly chosen from 30 grape clusters at 7:00 am every 14 days [14, 28, 42, 56, 70, 84 and 98 days after treatment (DAT)] for analysis of longitudinal diameter (LD) and transverse diameter (TD). [page 3, paragraph 2, and line 116-119].
|
||
|
Comments 8 The definition of Figures 2 and 3 is low; the symbols identifying each treatment cannot be visualized well, making it difficult to understand the graphs.
|
||
|
Response 8: The definition of Figures 2 and 3 is 300dpi. I suggest that Figures 2 and 3 could be enlarged when typesetting. We upload the Figures 2 and 3 along with the revised manuscript.
|
||
|
Comments 9 The representation of Figure 2C is not clear, as it shows photos of berries and clusters without explanation.
|
||
|
Response 9: Agree. Moreover, bunch compactness decreased in response to GA3 treatments (T2 and T3 treatments). the berry size was significantly higher in T1, T2, and T3 treatment groups than in the CK group (Figure 2C), which suggesting the expansion effect of GA3. [page 6, and line238-240].
|
||
|
Comments 10 In the analysis of berry sugar composition, they indicate the presence of traces of sucrose, but they do not present the data.This does not correspond to the phloem loading data, where the predominant sugar is sucrose. There is no explanation provided for this discrepancy. Moreover, this latter finding (the high amount of sucrose in the phloem flow) is in stark contrast to the study by Wang et al that describes the experimental setup, where sucrose is very low.
|
||
|
Response 10: Thank you for this comment. I really appreciate your serious attitude. Thank you very much! I’m sorry for this mistake. The unit of sucrose is ug/g for the phloem loading data. We upload the original figure as follows:
|
||
|
Comments 11 On more than one occasion, evolutions throughout shelf life are mentioned (Ln 281 & 290), but there is no mention of post-harvest analysis in the Materials and Methods section.
|
||
|
Response 11: The berries were harvested at 100 DAT, not consisting post-harvest analysis. The shelf life refers: the period of berries in the vines, the hanging life.
|
||
|
4. Response to Comments on the Quality of English Language |
||
|
Point 1: I am not qualified to assess the quality of English in this paper. |
||
|
Response 1: TopEdit (www.topeditsci.com) has polished this paper during the preparation of this manuscript. |
||
|
5. Additional clarifications |
||
|
No |
||
Round 3
Reviewer 2 Report
Comments and Suggestions for Authors
The authors answered all the questions.